# In Situ Synthesis of Gold Nanoparticles in Layer-by-Layer Polymeric Coatings for the Fabrication of Optical Fiber Sensors

**DOI:** 10.3390/polym14040776

**Published:** 2022-02-16

**Authors:** María Elena Martínez-Hernández, Javier Goicoechea, Pedro J. Rivero, Francisco J. Arregui

**Affiliations:** 1Electrical, Electronic and Communications Engineering Department, Arrosadia Campus, Public University of Navarre (UPNA), 31006 Pamplona, Spain; javier.goico@unavarra.es (J.G.); parregui@unavarra.es (F.J.A.); 2Institute of Smart Cities (ISC), Arrosadia Campus, Public University of Navarre (UPNA), 31006 Pamplona, Spain; 3Engineering Department, Campus de Arrosadía S/N, Public University of Navarre (UPNA), 31006 Pamplona, Spain; pedrojose.rivero@unavarra.es; 4Institute for Advanced Materials and Mathematics (INAMAT2), Public University of Navarre (UPNA), 31006 Pamplona, Spain; 5Navarra Institute for Health Research (IdiSNA), 31008 Pamplona, Spain

**Keywords:** polymers, optical fiber tip, gold nanoparticles, relative humidity, infrared, interferometric response, in situ synthesis

## Abstract

A new method is proposed to tune the interferometric response of wavelength-based optical fiber sensors. Using the nanoparticle in situ synthesis (ISS) technique, it is possible to synthesize gold nanoparticles (AuNPs) within a pre-existing polymeric thin film deposited at the end-face of an optical fiber. This post-process technique allows us to adjust the optical response of the device. The effect of the progressive synthesis of AuNPs upon polymeric film contributed to a remarkable optical contrast enhancement and a very high tuning capability of the reflection spectra in the visible and near-infrared region. The spectral response of the sensor to relative humidity (RH) variations was studied as a proof of concept. These results suggest that the ISS technique can be a useful tool for fiber optic sensor manufacturing.

## 1. Introduction

The great versatility of polymers has resulted in their extensive use in different sectors [1,2], offering outstanding mechanical, dielectric, thermal, and optical properties at a relatively low cost. They may be found in a variety of modern electronic device industries [3], such as aerospace transportation, 5G communication services [4], and 3D printing [5]. Moreover, another interesting field for their use is in the optical sensor field, where polymers have been used to create optical waveguides and special fibers [6]. Furthermore, a wide variety of optical sensitive coatings have been used for the detection of diverse variables, such as humidity, pH, temperature, electrical fields, chemicals, biological agents [7,8,9,10,11,12], and many others. The utilization of these polymeric materials has allowed us to obtain high precision sensors and improve the response of these sensors with better sensitivity and selectivity [13]. The versatility of polymers makes it possible to use them in different ways in sensitive coatings, whether playing an active role in the sensor [14,15] or acting as a support material for other sensitive agents, such as dyes [16], fluorophores [17], nanoparticles [18,19], etc. One interesting application of certain polymeric composites is that they can be used as a synthesis medium, enabling the In Situ Synthesis (ISS) of nanoparticles within the polymeric matrix using thermal treatments [20] or chemical reduction methods [21]. Compared to other techniques for the nanoparticle fabrication in optical coatings that may require special conditions or complicated equipment [22,23,24], the ISS technique is a wet-chemical synthesis route, capable of loading the nanoparticles gradually into previously existing optical coatings, under room conditions, while allowing for the continuous monitoring of optical responses by sensors. It is a simple and promising technique for nanoparticle synthesis and a good post-fabrication method to modify the optical properties of sensitive coatings.

Regarding possible optical sensor approaches, it is possible to find alternatives regarding intensity-based optical fiber sensors, through which sensed information is translated into an intensity-based signal. This approach has many advantages, including low cost, easy operation, and flexibility [25,26]. However, they do have some drawbacks, such as certain stability issues, sometimes related to variations in the intensity of the light sources, making it necessary to use additional referencing systems [27]. Alternatively, wavelength-based optical fiber sensors use changes in the wavelength as their detection principle [28]. Although they may be more complex and require more sophisticated optical instrumentation, they are considered more robust compared to intensity-based ones [29] in as far as their responses do not depend as much on the stability of light sources, etc. Some of the most used devices for wavelength-based approaches are fiber Bragg gratings (FBGs), tilted fiber Bragg gratings (TFBGs), long-period gratings (LPG), or interferometers. Some authors have reported on the use of Fabry–Perot interferometers (FPI), which operate by means of building thin films at the end-face of optical fibers [30]. This is a very simple interferometric approach that has many advantages as an optical sensor. For this kind of application, wavelength-based interferometric sensors can be read out with extremely high accuracy and, furthermore, it is immune to the instabilities which plague its contemporaries, such as optical systems, power levels, or polarization state [31].

This configuration is very compact and works using a reflective configuration that makes it possible to work with an optical sensing probe, or optrode [22,32,33,34,35,36,37]. As previously commented upon, in some approaches, the polymers work in support of active agents, such as luminescent materials or other elements [38]. Other schemes base their sensing mechanism on the response of the polymer itself which acts as the only sensing element, allowing us to detect changes in optical reflection [39]. Unfortunately, in these last cases, the FPI cavities show moderately low optical contrast on their interferometric reflection due to the similar refractive index of most of the polymers in relation to the fiber. In addition to this issue, another challenge is tuning the optical spectral response of the FPI to match its optimal active wavelength range with the optical instrumentation’s optimal window [40].

In this research, the in situ synthesis (ISS) technique is proposed as a post-fabrication method of immobilizing AuNPs within the optical fiber tip (FPI) cavity, just after the polymer film fabrication. This synthesis technique has been successfully used by other authors, such as Rubner and co-workers [41,42], as a powerful tool to synthesize nanoparticles inside some coatings, such as LbL coatins, sol-gel matrices, hydrogels, etc. With ISS, the AuNPs can be gradually loaded into a previously fabricated LbL coating at the end-face of an optical fiber, while the optical response of the FPI is continuously monitored. The purpose of the incorporation of these AuNPs is to enhance the contrast of the optical response of nano-FPI cavities and, at the same time, to tune, if possible, the optical spectral response of the nano-FPI.

## 2. Materials and Methods

### 2.1. Materials

The materials used to fabricate the polymeric multilayer thin film were poly (allylamine hydrochloride) (PAH) (Mw ~ 15,000), acting as polycation, and a poly (acrylic acid) (PAA) 35 wt% solution in water, acting as polyanion. In order to synthesize AuNPs, gold (III) chloride trihydrate (HAuCl_4_·3H_2_O) was used as a precursor. Metallic nanoparticles were reduced with a borane dimethylamine complex (DMAB). The pH value of the solutions was adjusted by adding a few drops of HCl or NaOH. All the chemical reagents were provided by Sigma-Aldrich (St. Louis, MO, USA). 

### 2.2. Optical Detection Setup

Optical fiber sensors were made from multimode optical fibers, 62.5/125 μm core, and cladding, respectively, with FC/PC-FC/PC connectors (provided by Telnet Redes Inteligentes S.A., Zaragoza, Spain). The sensor structure was fabricated on the perpendicular cut end of the optical fiber in a reflection setup. The sensor was characterized using a 2:1 multimode optical fiber coupler (see Figure 1). One of the connectors of the optical fiber coupler was connected to a halogen white source (model TAKHI from Pyroistech S.L., Pamplona, Spain). The other end was connected to another fiber coupler to collect the optical response in the visible and infrared region at the same time with a CCD spectrometer (model HR4000-UV from Ocean Optics, Dunedin, FL, USA) and a NIR-Quest (Ocean Insight, Hakuto Singapore Pte Ltd., Singapore).

### 2.3. In Situ Synthesis (ISS) of Gold Nanoparticles into Polymeric Layer-by-Layer (LbL) Films

The AuNPs are synthesized inside an existing polyelectrolyte coating, using this solid structure as the stabilization medium where the AuNPs are going to be loaded. Therefore, the fabrication process of the sensing coatings has been performed in two steps (Figure 1). First (Step A), a polymeric coating was fabricated using the layer-by-layer assembly technique (LbL) by dipping the substrates into a sequence of oppositely charged polyelectrolyte solutions. Some experimental parameters, such as polyelectrolyte concentration, pH, and ionic strength strongly affect the properties of LbL coatings [43]. In this work, a solution of PAH (10 mM) was used as a polycationic solution, and a PAA solution (10 mM) was used as a polyanion. The optical fiber substrates were immersed in each charged solution for 5 min. After every polyelectrolyte adsorption step, it is necessary to rinse the assembled tools in ultrapure water with the same pH as the polyelectrolytes [44,45]. All solutions were adjusted to pH = 4. Before starting the deposition of layer-by-layer, the end of the optical fiber was immersed in KOH (1 M) for half an hour to achieve substrate surface electrostatic charge.

Once the LbL coating that is going to act as the synthesis medium has been built-up, the AuNPs are synthesized within. In this new process (Step B), the previous polymeric matrix obtained by the LbL assembly was immersed in an aqueous solution of gold (III) chloride trihydrate (HAuCl_4_·3H_2_O) at room temperature for 5 min, removed, and rinsed with ultrapure water. During this step, some Au^3+^ cations were immobilized in the charged groups present in the LbL polyelectrolytes. Afterward, those gold ions loaded (L) into LbL polymeric coating have been reduced (R) by dipping again the LbL coatings in a dimethylamine borane complex solution (DMAB 0.1 M) at room temperature for 5 min and rinsed with ultrapure water. This reducing agent (DMAB) makes possible the in situ chemical reduction of the gold cations (Au^3+^) to gold nanoparticles (Au^0^) [46]. This process can be repeated as many times as desired, taking into account that each cycle is referred to as a load/reduction (L/R) cycle. With every L/R cycle, new Au^3+^ ions are loaded into the LbL thin film, and further reduced by the DMAB, contributing to the creation of new AuNPs, as well as growing previously existing ones. The resultant AuNP loaded LbL films were stable under the normal conditions of any other layer-by-layer coating, and no evident degradation of the coatings was observed during the experiments.

## 3. Results

### 3.1. Layer-by-Layer and AuNPs In Situ Synthesis Characterization

Originally, before this investigation into the effects of sensing material upon optical fiber, coatings were fabricated and placed on glass slides as substrates in order to characterize the sensing material and to determine the progressive AuNPs’ load through the analysis of the localized surface plasmon resonance (LSPR) peak of the in situ synthesized nanoparticles in the LbL films. Thanks to this characterization, it was possible to analyze the conditions necessary for obtaining an intense optical absorption band at 550 nm, typical of AuNPs [47]. The synthesis of AuNPs in the LbL films was carried out with different numbers of (PAH/PAA) bilayers and loading/reduction (L/R) cycles. After some previous experimental work, LbL films of 15 and 25 (PAH/PAA) bilayers were selected for analysis. Additionally, the ISS of AuNPs into the LbL films has been carried out using a number of different L/R cycles.

The UV-VIS spectra can be observed in Figure 2, where the UV-VIS absorption spectra of different LbL+ISS films are shown. It may be appreciated, even with the naked eye, that substrates with 25 bilayers of PAH/PAA present a stronger violet coloration compared to the samples with 15 bilayers of PAH/PAA (with the same number of L/R cycles). The absorption values shown in Figure 2 corroborate this fact. It is appreciable that with every L/R cycle, not only does absorption increase, but it also incurs a slight redshift of the LSPR peak due to the aggregation of AuNPs within the LbL film. The LSPR center peak was localized near 528 nm for the second cycle of loading/reduction, ${\left(L/R\right)}_{2}$, and a small redshift of the LSPR peak was observed, reaching a maximum of 552 nm for ${\left(L/R\right)}_{5}$.

The results of the spectra show that the number of PAH/PAA bilayers affects the absorbance values obtained from the Au-LSPR peak with a subsequent analysis of AuNPs by L/R. When a higher number of L/R cycles has been performed, a more substantial violet coloration in the LbL films can be observed in both cases, for 15 and 25 bilayers of (PAH/PAA). The absorbance values of the coatings with 25 bilayers of PAH/PAA are slightly higher than those of the coatings with 15 bilayers of PAH/PAA. Additionally, in both cases, the L/R cycles evidence the same results: a strong LSPR peak and an appreciable violet coloration obtained via the glass substrate. Since the results are similar for both 15 and 25 bilayers of PAH/PAA, in order to optimize experiments hereafter, it is recommended that all subsequent experiments use polymeric coatings of 15 bilayers.

### 3.2. Characterization of Sensitive Film

Once the films have been characterized on microscope glass slides, LbL+L/R coatings are fabricated on perpendicular cut ends of optical fiber and their reflectivity is studied (see Figure 1). In this case, the end face of the optical fiber is the sensitive region where the LbL polymeric film is deposited and, later, the AuNPs are synthesized using the ISS technique. The same process used in Section 3.1 was applied to the sensitive region to deposit ${\left(\mathrm{PAH}/\mathrm{PAA}\right)}_{15}\;$+ ${\left(L/R\right)}_{m}$ (m = 1, 2, 3, 4) and the optical response was registered.

Figure 3A shows the reflectivity, due to the FPI, constituted by the LbL ${\left(\mathrm{PAH}/\mathrm{PAA}\right)}_{15}$ thin film without any AuNPs. Using halogen white light sources, it is possible to observe a Fabry–Perot interferometric response, despite the low coherence of the optical source used in this experiment [48]. It is known that any variation in physical or optical characteristics of the polymeric thin film will induce changes in the optical response on the FPI cavity, opening the door to the development of optical fiber sensors. However, if we only use polymeric thin films, we obtain an interferometric response with a low contrast due to the similar refractive indexes of the polymeric film and the fiber.

The next step is loading the thin film with AuNPs by means of ISS. Figure 3B shows spectra evolution with every L/R cycle. In this figure, it is possible to observe the interaction between two phenomena. First, the formation of a reflection peak in 580 nm is due to the incorporation of AuNPs. Second, the interferometric redshift with an improvement in the contrast of the optical response of nano-FPI cavities. Gouva et at. have reported similar reflectivity responses in AuNP decorated optical fiber end faces [49]. In their work, the authors also assume that the optical response is due to both the interferometric response of the layer and the LSPR optical absorption of the AuNPs [50]. In the results shown in Figure 3B, it is possible to observe that, with every L/R cycle, a reflectivity of around 560–580 nm was strongly increased, compared to the original LbL polymeric thin film.

Of special interest is the behavior of every L/R cycle (Figure 3B), apart from LSPR-based reflectivity, increasing because there is a redshifted interferometric response. In the first L/R cycle, a slight bending in the reflectivity curve around 740 nm can be observed, suggesting a light interferometric maximum, and a minimum reflection of around 540 nm. In the (L/R)_2_, while the LSPR is not noticeable, the minimum reflection redshifts to 560 nm, while the maximum goes beyond the scope of the spectrometer (800 nm). In L/R cycles 3 and 4, the local minimum reflectivity continues its shift to 690 nm and 720 nm, respectively, and both AuNPs-LSPR and FPI interferometry phenomena can be observed at the same time.

Additionally, in Figure 3C it is possible to appreciate the improvement in the contrast of the interferometric signal (visibility enhancement) after the use of ISS. More specifically, for the (L/R)_4_ coating, the interferometric contrast was enhanced fourfold with respect to the plain polymeric coating, from 16.8% to 70% ΔR. The LSPR center peak was localized near 577 nm (Figure 3C).

### 3.3. Optical Fiber Sensor Characterization

With the objective of studying a proof-of-concept case, since the polymeric PAH/PAA coating is sensitive to humidity [51], devices with plain PAH/PAA films and devices with an enhanced spectral response by means of the ISS technique are characterized against relative humidity (RH) variations. Five sensors were fabricated, from ${\left(\mathrm{PAH}/\mathrm{PAA}\right)}_{15}\;$+ ${\left(L/R\right)}_{0}$, or plain PAH/PAA coatings, to ${\left(\mathrm{PAH}/\mathrm{PAA}\right)}_{15}\;$+$\;{\left(L/R\right)}_{4}$, all under the same conditions, and every sensor was exposed to the same cycles of humidity for the same amount of time in a climatic chamber. The temperature in the climatic chamber was set at 20 °C during all the experiments in order to avoid complications due to additional parameters.

The sensor with plain LbL polymeric thin film and no AuNPs (Figure 4A) shows a reflective change of 2.86% for values of RH from 20 to 80%, with low contrast and no noticeable wavelength change. For the sensors with 1 and 2 L/R cycles, the spectral shape of the reflectivity response started to change, but still, no noticeable wavelength changes were registered (Figure 4B,C), for 20–80% RH variations. With every L/R cycle, an improvement of the interferometric response contrast was appreciable. Nevertheless, it is only from the ${\left(L/R\right)}_{3}$ cycle that it is possible to observe a clear reflective AuNPs-LSPR peak near 565 nm (Figure 4D), and a spectral response with peaks and valleys also spectrally sensitive to RH changes; a redshift with a minimum reflection around 760 nm. Furthermore, in Figure 4E, it is shown that the response of a ${\left(L/R\right)}_{4}$ cycle sensor, is to exhibit an appreciable redshift of 30 nm of the interferometric response when increasing RH from 20% to 80%. A higher number of L/R cycles move the interferometric minimum to wavelengths beyond the limits of the spectrometer used in this experimental setup. Thus, in this case, ${\left(L/R\right)}_{4}$ devices showed the optimal optical response. The optical responses shown in Figure 4 were registered in a climatic chamber using repetitive 20–80% RH cycles, and all of them showed good repeatability. One example of the sensors’ repeatability is shown in Figure 5A, where the response shows very similar behavior when submitted to four RH cycles in a climatic chamber, with no significative drift.

The wavelength shift of the minimum value of the interferometric response of the sensor ${\left(\mathrm{PAH}/\mathrm{PAA}\right)}_{15}$ + ${\left(L/R\right)}_{4}$ was analyzed in the climatic chamber with 4 cycles from 20% to 80% of RH at 20 °C to study the dynamic response of the sensor. In Figure 5A it is appreciable that the response of the sensor follows the evolution of the RH inside the climatic chamber and each RH 20%–80%–20% cycle is registered as a wavelength shift of approximately 9 nm in the sensor response.

The average RH rise (Figure 5B) and fall (Figure 5C) provide repetitive sensor behavior with low hysteresis. The response time is limited by the climatic chamber. The slight difference in rise and fall responses can be attributed mainly to the use of polyelectrolytes like PAH and PAA that exhibit swelling/deswelling hysteresis in humid air environments [52]. In the case of the RH fall (Figure 5C), the response of the sensors was very linear.

### 3.4. Effects of the Number of L/R Cycles on the Optical Fiber

One of the goals of this work is to demonstrate the use of the ISS technique as a post-fabrication tool for tuning the optical properties of polymeric sensing coatings. In this section, the amount of L/R cycles under study have been extended significantly in order to study if it is possible to extend the appreciable interferometric response of the sensor to higher wavelengths of visible and infrared regions. Figure 6 shows the reflectivity spectra and the shift of the spectrometric response towards the infrared region with the increase in the number of L/R cycles.

In Figure 6 different sensor responses are represented, with L/R cycles varying from 1 to 20. It is apparent that two different phenomena coexist. Firstly, the LSPR-induced reflectivity peak in the 550–580 nm band. Secondly, the interferometric behavior of the AuNP-loaded LbL thin film. The presence of the AuNPs was critical to significantly enhance the contrast in the interferometric optical response. The first 8 L/R cycles demonstrated the build-up of the LSPR band and the initial displacement of the interferometric minimum to higher wavelengths. There, the following maximum and the high-reflectivity LSPR band still overlapped. Nevertheless, from cycle 12 on, the next interferometric maximum goes apart from the LSPR band, appearing clearly visible. The increase in the AuNPs loaded into the LbL thin film allows us to move the interferometric response to the near-infrared region in a controlled way, so the spectral response of the sensor can be adjusted to the desired wavelength window. In the L/R cycle 20, the minimum of the interferometric response is centered at 1340 nm. Figure 7 shows the evolution of the FPI minima as the number of L/R cycles were increased. As observed, there is a strong linear dependence between the redshift of the FPI minimum and the amount of AuNPs loaded into the coating.

A video has been provided as Appendix A where ${\left(L/R\right)}_{30}$ cycles were registered, showing that with every L/R cycle the interferometric quality factor improves and moves to infrared wavelengths.

## 4. Conclusions

In this investigation, the ISS technique has been presented as a tool for the incorporation of AuNPs into previously fabricated layer-by-layer polymeric sensing coatings. In this case, the coatings under study were deposited at the end-face of optical fibers to fabricate nano-FPI cavities. The ISS allowed us to tune the interferometric response of the nano-FPI and improve the contrast of the optical response of the device.

The optimal fabrication condition was studied via a previous characterization of polymeric coatings and ISS-AuNPs on glass substrates. Subsequently, combinations of ${\left(\mathrm{PAH}/\mathrm{PAA}\right)}_{15}$ with different (L/R) cycles were tested on optical fiber to achieve a higher optical contrast of the interferometric response (visibility) with a fourfold enhancement. The devices’ response to relative humidity was studied as a proof of concept of the application of the ISS technique to improve the sensing characteristics of the sensors.

Furthermore, the ISS is a powerful post-process technique to adjust the optical properties of a polymeric thin film, which is especially useful in interferometric coatings. With the addition of more L/R cycles, it was possible to tune the interferometric minimum reflection valley from the VIS (600 nm) to NIR (more than 1400 nm). This opens up a new possibility; to work with wavelength-based sensors that are capable of adjusting interferometric parameters.

## Figures and Tables

**Figure 1 polymers-14-00776-f001:**
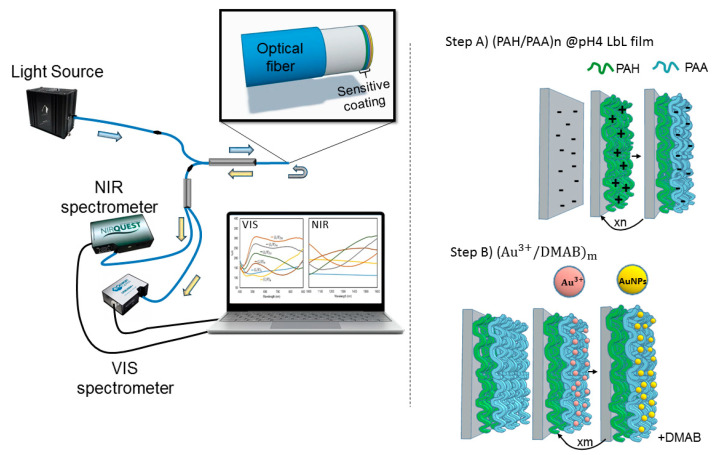
On the left, experimental setup. The blue arrows represent the light launched at the device (schematically depicted at the zoom view), the grey arrow is the light reflected at the device and the yellow arrows represent the reflected light that reaches the spectrometers. On the right, schematic illustration of the layer-by-layer nano-assembly built-up (Step A) and a further in situ synthesis (ISS) of the gold nanoparticles into the previously created LbL films (Step B). The construction of the sensitive coating ${\left(\mathrm{PAH/PAA}\right)}_{n}$ + ${\left({\mathrm{Au}}^{3+}/\mathrm{DMAB}\right)}_{m}$ was carried out on the end face of a 62.5/125 μm multimode optical fiber (zoom view on the right).

**Figure 2 polymers-14-00776-f002:**
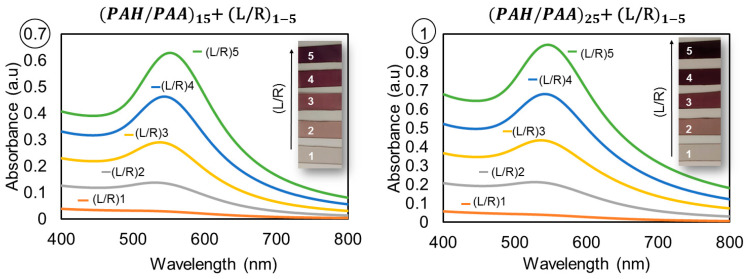
UV-VIS spectra of the in situ synthesis (ISS) of the gold nanoparticles (AuNPs) into LbL films for different numbers of PAH/PAA bilayers (15 and 25) and for 1 to 5 L/R cycles. Inset, photographs of the microscope glass slide.

**Figure 3 polymers-14-00776-f003:**
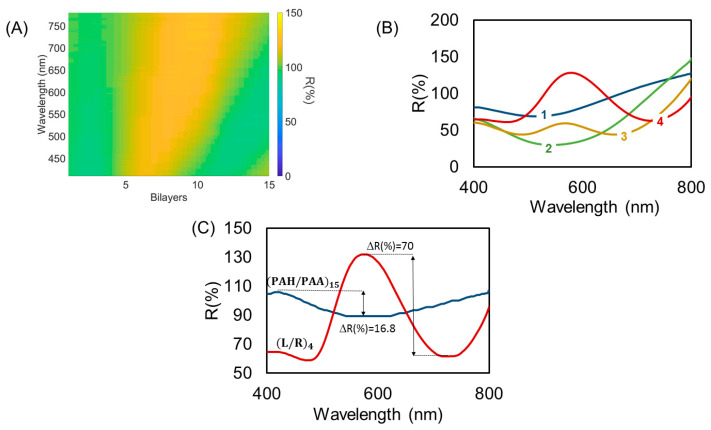
(**A**) Evolution of the spectral characteristic of the reflected light due to white light interfering with the LbL-${\left(\mathrm{PAH}/\mathrm{PAA}\right)}_{15}$ thin film. (**B**) Optical reflectance of the LbL films when ISS of gold nanoparticles is performed, from 1 to 5 L/R cycles. (**C**) Optical reflectance comparison between LbL-${\left(\mathrm{PAH}/\mathrm{PAA}\right)}_{15}$ thin film (without AuNP) and ${\left(\mathrm{PAH}/\mathrm{PAA}\right)}_{15}\;$+$\;{\left(L/R\right)}_{4}$.

**Figure 4 polymers-14-00776-f004:**
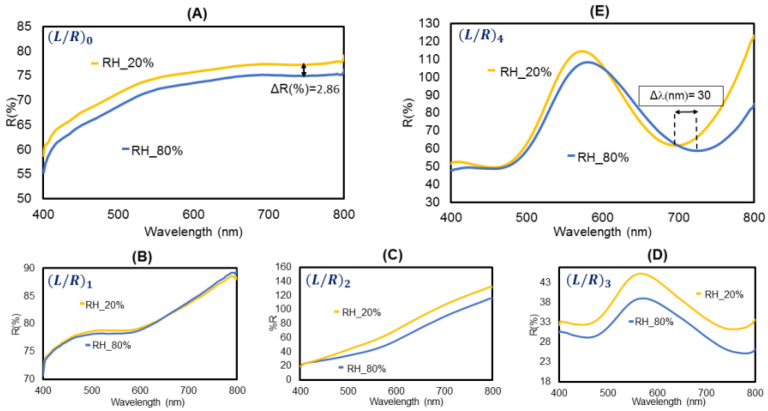
Spectral response of (**A**) LbL-${\left(\mathrm{PAH}/\mathrm{PAA}\right)}_{15}$; (**B**) ${\left(\mathrm{PAH}/\mathrm{PAA}\right)}_{15}$ + ${\left(L/R\right)}_{1}$; (**C**) ${\left(\mathrm{PAH}/\mathrm{PAA}\right)}_{15}\;$+ ${\left(L/R\right)}_{2}$; (**D**) ${\left(\mathrm{PAH}/\mathrm{PAA}\right)}_{15}\;$+ ${\left(L/R\right)}_{3}$; (**E**) ${\left(\mathrm{PAH}/\mathrm{PAA}\right)}_{15}\;$+$\;{\left(L/R\right)}_{4}$ thin film fabricated onto the end of the fiber tip for 20 and 80% values of relative humidity (RH).

**Figure 5 polymers-14-00776-f005:**
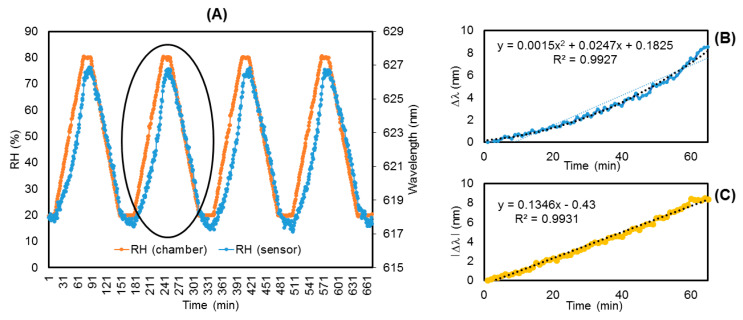
(**A**) Shift wavelength interferometric response of ${\left(\mathrm{PAH}/\mathrm{PAA}\right)}_{15}\;$+ ${\left(L/R\right)}_{4}$ with respect to cycles of RH from 20 to 80% in climatic chamber. (**B**) Response of the second RH cycle in climatic chamber in the rise interval. (**C**) Response of the second RH cycle in climatic chamber in the fall interval.

**Figure 6 polymers-14-00776-f006:**
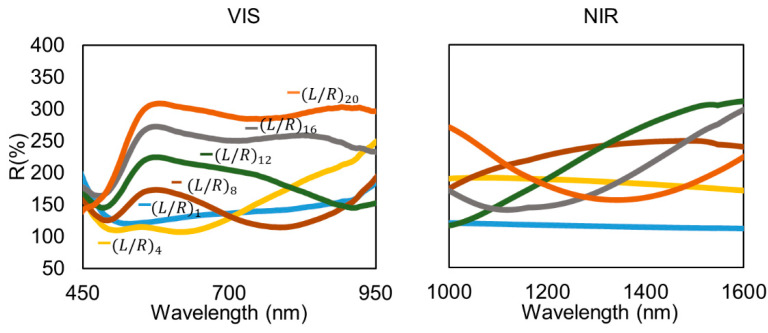
A demonstration of the interferometric response of the sensor in the visible and infrared region, increasing the L/R cycles.

**Figure 7 polymers-14-00776-f007:**
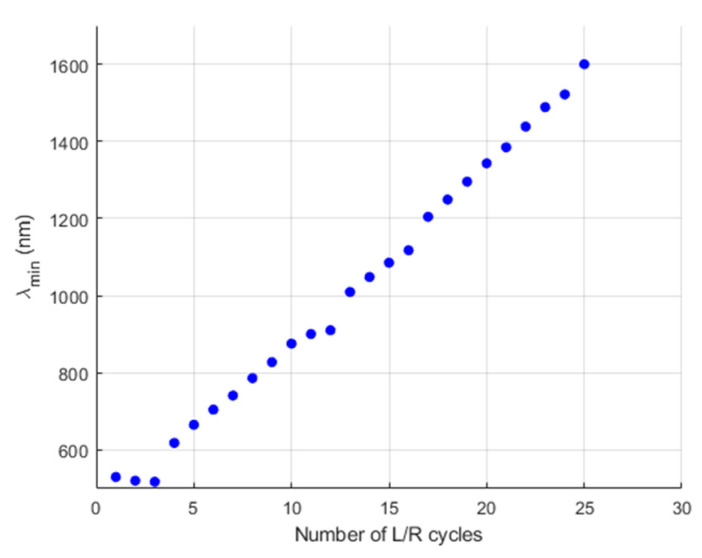
Evolution of the wavelength of the minimum interferometric response from the Fabry–Perot optical fiber interferometers and the number of L/R cycles carried out during its fabrication.

## Data Availability

Not applicable.

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
