# Peer review of "In Situ Synthesis of Gold Nanoparticles in Layer-by-Layer Polymeric Coatings for the Fabrication of Optical Fiber Sensors"

_polymers, 2022, doi:10.3390/polym14040776_

Round 1

Reviewer 1 Report

Francisco J. Arregui group has developed a new synthetic way for In-Situ Synthesis of Gold Nanoparticles in Layer-by-Layer 
Polymeric Coatings for the Fabrication of Optical Fiber Sensors, in this article authors, has discussed the interferometric response of wavelength-based optical fibre sensors, Author should include some related information in the introduction that will enhance the readers understanding and improve the visibility of the articles. I would like to suggest some of the relevant articles that should be cited in the introduction part

Gold nanoparticles promoted the formation and biological properties of injectable hydrogels. Biomacromolecules21(9), 3782-3794.

The author should discuss the stability of nanoparticle and polymer coat over the nanoparticle ( how much % of polymer coated on the nanoparticle w/w ratio) this information is very important for this article.

Author Response

We would like to thank to the anonymous reviewer for his/her comments. All the changes have been highlighted in yellow for a better understanding and localization of them, as it can be appreciated in the revised version of the manuscript. Finally, we hope that this new revised version of the manuscript can be published in Polymers-MDPI.

Reviewer 2 Report

This paper reports a new method to tune the interferometric response of wavelength-based optical fiber sensors using the NPs in ISS technique, where  a pre-existing polymeric thin film deposited at the end-face of an optical fiber is used. I have some comments.

  1. when is mentioned that the wavelength based sensors are more robust compared to the intensity-based ones, I miss more discussion and references since depend of the applications and some critical points that from what we want. Please check this literature and re-write this sentence since we can need a low-cost solution and intensity schemes are the best in many points like presented in: Optical Fiber Technology 41, 205-211, 2018; Advanced Photonics Research 2 (8), 2100044, 2021.
  2. How about the complex and cost associated with this proposed technique compared with the ones in the literature?
  3. Can we have a good repeatability of the performance for systematic cycles (Fig. 4)? Please comment.
  4. How about the reproducibility of the proposed? Can we achieve identical probes? 
  5. How can we improve the difference in Fig. 6? How about the decreasing?

Author Response

(The authors gave the same response as above.)

Round 2

Reviewer 2 Report

The paper is ready for publication